# Corporate Philanthropy Strategy and Sustainable Development Goals

**Hui-Cheng Yu** [1,*] **and Lopin Kuo** [2]

1. Department of Accounting Information, Chihlee University of Technology, New Taipei City 22050, Taiwan
2. Department of Accounting, Tamkang University, New Taipei City 25137, Taiwan; lopinkuo@gmail.com
* Correspondence: timmy0927@yahoo.com.tw

**Abstract:** This paper investigates the charitable giving of Chinese firms from the perspectives of four sustainable development goals (SDGs), including *Economy*, *Operation*, *Harmony*, and *Management*. By converting corporate financial data into four independent variables, namely *Economy*, *Operation*, *Harmony*, and *Management*, this study explores philanthropic giving for SDGs. The empirical evidence shows that corporate philanthropy has a significant and positive effect on *Economy*, *Operation* and *Harmony*, and it is negatively related to *Management*. This study finds that the sample firms would undertake some social responsibilities for the economic and political benefits of legitimization or corporate philanthropy.

**Keywords:** philanthropic giving; sustainable development goals; social responsibility

## 1. Introduction

The past economic reforms in China helped create economic growth and wealth, but they also widened the gap between rich and poor [1]. This widening gap has become a serious barrier to social harmony and the fulfillment of sustainable development goals (SDGs) in China [2]. The fundamental goal of sustainable development is to ensure that the environment, economy, and society are developing toward sustainable prosperity. The Chinese government has noticed the severity of this problem and added a poverty reduction program in the Thirteenth Five-Year Plan for Economic and Social Development (the poverty issue was not addressed in previous plans for economic and social development).

In order to promote poverty reduction collaboration between the eastern and western regions, companies, social organizations, and individuals are encouraged to support and participate in poverty reduction to achieve the goals of social harmony and sustainable development [3].

As far as narrowing the gap between rich and poor is concerned, corporate philanthropy is a feasible measure. However, with economic reforms, the traditional Chinese culture has also evolved. For example, traditionally, it was considered better to keep a low profile when doing a good deed (e.g., making a charitable donation). Today, this tradition has changed quietly in the Chinese society probably because the media has more interest in reporting the good deeds of individual or corporate donors. There is probably a social belief that corporate donations can help create a positive corporate image or a better relationship with stakeholders [4]. It is also possible that corporate philanthropic giving is considered the "utmost fulfillment of corporate social responsibility" [5] or an indicator of social performance of the firm [6]. Recently, corporate philanthropy is recognized as one of the fundamental factors of "corporate citizenship" [7]. This is because sustainable development is an important aspect of business strategies [5].

Studies have shown that well-planned and implemented corporate social responsibility (CSR) activities not only contribute to corporate sustainability [8] but also play a pivotal role in the development of a competitive advantage [9] and in the improvement of financial performance [10–13]. However, previous research offers limited insights into the SDGs of

Chinese firms. Understanding how firms and industries react to different SDGs is one of the first steps to studying corporate philanthropic strategies.

Based on the above ideas, the focus of this article is placed on how the philanthropic strategy of businesses promotes the achievement of sustainable development goals. The research concept is shown in Figure 1.

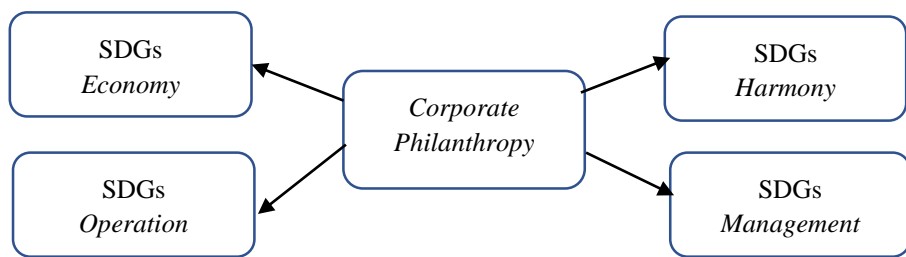

**Figure 1.** Research concept.

The main purpose of this study is to examine the impact of corporate philanthropy on the sustainability development goals of Chinese listed companies by drawing on a sample from the China Stock Market and Accounting Research (CSMAR) database. CSMAR is a very well-known database in China, and many academic studies employ data from this database. The database contains a collection of financial data of publicly listed firms in China and macroeconomic statistics of China.

This study has a number of contributions. First and most important of all, this study adopts a research design that classifies sample data by disaster period and firms by industry characteristic to examine if there are differences in the SDGs between disaster periods and non-disaster periods and between controversial industries and non-controversial industries. The results show no significant difference in the SDGs between disaster and non-disaster periods or between industry types. To be more specific, the sample firms have strategic considerations for the SDGs, but their considerations are not significantly affected by the occurrence of a disaster or industry characteristics. This empirical evidence is not a common finding, especially from emerging economies. Second, most research of CSR in China uses state ownership [4,14,15] as a proxy for political interest. In this paper, government subsidy is used as a proxy variable for the SDGs of the political interest (i.e., operation), because this variable can better reflect practical conditions. The results reveal that the SDGs of the operation is affected by the firms' philanthropic giving. Third, previous research indicates that harmony is one of the SDGs and seldom uses financial statistics to represent this variable. In this study, net sales is used as a dummy variable to proxy for the SDGs of the altruistic (i.e., harmony). The results show that the sample firms achieve the harmony as the sustainable development goal. Finally, although managerial utility is one of the SDGs but prior research seldom uses financial data to represent this variable. This study uses CEO ownership to proxy for managerial utility (i.e., management). The empirical results indicate that the SDGs of the management is not motivated by the firms' philanthropic giving.

## 2. Literature Review and Hypotheses Development

### 2.1. Chinese Enterprise Charity

In light of the economic growth and wealth brought by economic reforms, enterprises in China have shifted their primary focus to economic benefits, resulting in an imbalance between profits and social welfare [16]. To some extent, economic reforms have brought adverse effects on traditional social welfare values [16].

The shift of business focus to economic benefits has widened the gap between rich and poor and intensified social conflicts, which will in turn seriously hinder China's pursuit of sustainable development goals and social harmony [2]. Charitable donation can be one of the most effective means to reduce these pressures. In fact, probably influenced by the

traditional Chinese idea of "Contribute to the society with whatever we have taken from it", enterprises have long been the main providers of philanthropic donations in China.

Chinese companies are used to viewing donations as costs or burdens [17]. In terms of philanthropic contributions, empirical studies [18] have shown that the state-owned enterprises are inferior to privately-owned enterprises (POEs). Hence, the institutionalization of philanthropy policies will be one of the major tasks of the government.

## 2.2. Legitimacy Theory Perspective

The concept of legitimacy theory is that organizations are continually seeking to ensure that they operate within the bounds and norms of their respective societies [19]. According to legitimacy theory, legitimacy can be manipulated by an organization to certain extent [20,21]. It is also believed that an organization can also make strategic decisions to change their status of legitimacy and build a good foundation of resources through proper corporate behavior that can adapt to their activities and constantly changing social opinions. In other words, an organization can use various means (e.g., philanthropic donations) to ensure the legitimacy of its behavior.

Legitimacy theory assumes that organizations will keep looking for barriers and specifications that make them distinguished from others through operations [19]. Organizations are judged as legitimate when they establish a congruence between the social values associated with their activities and the norms of acceptable behavior in the social system of which they are a part [22]. Previous research [23] views CSR as an effective strategy of legitimization because enterprises can use signals of legitimacy or other symbolic actions to deliver a socially acceptable image [22] and convey their achievement of SDGs.

## 2.3. Goals for Corporate Charitable Strategy

The benefits of corporate philanthropy for a company are determined by the motivation behind the philanthropic behavior. Campbell et al. [24] have identified the main motivations for corporate philanthropic giving for the SDGs, including economic, political, altruistic, and managerial utility. Specifically, corporate philanthropy is a strategic activity with certain purposes [25], but the strategy behind the activity varies across enterprises. Therefore, based on Campbell et al. [24], this study will focus on four SDGs of corporate philanthropy, including the SDGs of profit maximization (i.e., economy), the SDGs of political interest (i.e., operation), the SDGs of altruistic (i.e., harmony), and the SDGs of managerial utility (i.e., management) as follows.

### 2.3.1. The SDGs of the Profit Maximization

Economy is one of the major goals of corporate sustainable development. It is widely agreed among enterprises that profit is the true goal of giving. Enterprises adopt this strategy to achieve either sustainable development or profit improvement [26,27]. In this case, philanthropic giving is used as a certain form of public relations or marketing advertisement [25] that can increase market demand for the company's products and decrease the price sensitivity of the company's products in the market [28]. Philanthropic activities can also mitigate the effects of labor cost. For instance, employee volunteering programs can improve staff morale, thus lowering staff turnover rates [27]. Moreover, philanthropic activities also have a positive effect on the economic performance of a firm [18,29]. However, recent studies [30] find that economic profit is not the main incentive for corporate philanthropy in China. Based on the above discussion, this study proposes the following hypothesis:

**Hypothesis 1.** *A firm's philanthropic giving can positively influence the SDGs of the economy.*

### 2.3.2. The SDGs of the Political Interest

According to Sánchez [31], operation is also an SDG for enterprises. Enterprises may engage in philanthropic activities for preservation of their corporate power and autonomy,

legitimization, and protection of their operated power. This view suggests that the strategy of corporate philanthropy is to maximize a company's operated interest. Neiheisel [32] reveals in an extensive study that enterprises will use charitable activities to enhance their legitimacy. In his conclusion, he mentions that corporate philanthropic giving is not intended to solve these social problems but to legitimize corporate power and protect the company from external threats. This conclusion implies that enterprises are supported by the society and stakeholders to use giving as a political approach to maintain or gain legitimacy [32]. In China, political interest may be a factor that motivates firms to engage in charitable donations [4,18,30]. Therefore, this study proposes:

**Hypothesis 2.** *A firm's philanthropic giving can positively influence the SDGs of the operation.*

### 2.3.3. The SDGs of the Altruistic

Harmony involves the intention to "do good". The basic concept of harmony is to be a good citizen, take maximization of public welfare as the company's reasonability and expect no return. This view is widely recognized among previous studies [31–35]. It is also conceptually similar to a socialist value in China—firms shall fulfill the goal of "enterprise-run societies" regardless of profitability. However, after economic reforms, many Chinese enterprises have aligned their corporate goals with the SDGs of the economy. The "enterprise-run societies" system is thus being seriously challenged. Therefore, this study proposes:

**Hypothesis 3.** *A firm's philanthropic giving can positively influence the SDGs of the harmony.*

### 2.3.4. The SDGs of the Managerial Utility

From the perspective of agency theory, corporate philanthropy can lead to an increase of the manager's self-interest but not maximization of shareholder wealth [36]. Navarro [27], Arulampalam and Stoneham [37], and Haley [35] all suggest that managers responsible for a firm's giving are likely to use their position of power to support causes they have personal empathy or affiliation with in contrast to causes that will benefit the interests of the firm. Philanthropic giving is a decision that can be made at the manager's own discretion [38]. In other words, personal values (e.g., the personal reputation of managers in the labor market) coupled with individual manager discretion are what essentially drive corporate giving programs. Moreover, previous research [27,39] has confirmed the positive relationship between agency cost and amount of donation. In other words, a firm is more likely to engage in charitable giving when its CEO has no ownership of the firm. This is why Navarro [27] defines this motive as "utility maximization and managerial discretion". Therefore, this study proposes the following hypothesis:

**Hypothesis 4.** *A firm's philanthropic giving can positively influence the SDGs of the management.*

## 3. Research Design
### 3.1. Sample and Method

The philanthropic giving and financial data of the sample firms were obtained from the China Stock Market and Accounting Research (CSMAR) database. The dataset comprised a total of 1027 observations during 2008–2015. Excluding missing values and firms in the financial and insurance industry, 960 observations remained. Considering the unique industry characteristics and capital structure of the financial and insurance industry, this study excluded all the observations from this industry.

### 3.2. Research Model

To identify Chinese firms' strategic motives for corporate philanthropy, the empirical model of this study can be expressed as follows.

$$SDGs_{i,t} = \alpha_0 + \beta_1 Donat_{i,t} + \beta_2 Idu_{i,t} + \beta_3 Disas_{i,t} + \beta_4\, Size_{i,t} + \beta_5\, Lev_{i,t} + \mu_{i,t} \tag{1}$$

where $i$ = 1, 2, . . . , 960 and $t$ = 2008–2015. *SDGs* denotes sustainable development goals in four aspects, including economy, operation, harmony and management. *Economy* is a dummy variable for the *SDGs* of the economy and is proxied by return on equity (ROE). In this variable, 1 represents a positive ROE value and 0 denotes a negative one. Authors use ROE to proxy for this variable because firms operate for the goal of maximizing shareholder wealth, and ROE is a measure that can adequately reflect the profit for shareholders. *Operation* is a dummy variable for the *SDGs* of the operation. It is proxied by government subsidies. In this variable, 1 denotes the firm has received government subsidies, and 0 denotes the firm has not. As compared with state-owned enterprise, using government subsidies as a proxy variable for political interest is more direct and can more effectively reflect the effect of the operation on enterprises. *Harmony* is a dummy variable for the *SDGs* of the harmony. It is proxied by net income. In this variable, 1 represents that the firm has a negative net sales, and 0 denotes that the firm does not. Authors use net sales to proxy for this goal because harmony suggests no expectation of return from engagement in CSR activities, and engaging in charitable giving regardless of profitability can be viewed as an altruistic behavior. *Management* is a dummy variable for the *SDGs* of the management utility. It is proxied by chief executive officer (CEO) ownership. In this variable, 1 denotes no ownership of the firm, and 0 denotes otherwise. When a CEO is a non-shareholding stakeholder and has the company's decision-making power, the CEO is more likely to use his/her discretionary power to engage in charitable activities in pursuit of self-interest [40]. In other words, the CEO may allocate corporate resources to CSR activities (e.g., donation) that do not contribute to an increase of firm wealth but only serve his/her self-interest (e.g., reputation) [41]. Therefore, it can be inferred that when the CEO does not own the company, the CEO is more likely to adopt the *SDGs* of the managerial utility, which is to use his/her discretionary power to engage in corporate philanthropy.

*Donat* denotes the amount of charitable donations. In order to rectify non-normality, the value of this variable will be converted into a natural logarithm.

*Idu* is as a dummy variable. This variable is 1 if the enterprise belongs to any of controversial industries (i.e., that are commonly perceived as more controversial in terms of the products or the environments including alcohol, tobacco, gambling, and weapons, adult entertainment, nuclear, oil, cement, and biotech, etc.); it is 0 if otherwise.

*Disas* is a dummy variable for disaster year. China suffered a catastrophic earthquake and winter storms in 2008 and 2009. This variable is 1 if the observation belongs to year 2008 or 2009; it is 0 if otherwise.

*Size* denotes the firm size. It is measured by the total asset of the company. *Lev* denotes debt ratio. It is measured by total debt over total asset.

## 4. Estimation Results

### 4.1. Descriptive Statistics

Table 1 shows the descriptive statistics of each variable. As shown in this table, the SDGs of the political interest (*Economy*) has a mean of 0.92, meaning that about 92% of the sample firms have received government subsidies. The SDGs of the managerial utility (*Management*) has a mean of 0.25. This reveals that about 25% of the sample firms are run by a CEO without ownership of the firm. In other words, in most sample firms, the CEO holds a certain amount of shares of the company. Philanthropic giving (*Donat*), after conversion into natural logarithms, has a mean value of 1.81, with a max of 5.73 and a min of 0.03. This large max-min difference shows that the amount of donation varies greatly among the sample firms. Controversial industry (*Idu*) is a dummy variable. The mean value of this variable is 0.13, suggesting that about 14% of the sample firms belong to one of the controversial industries.

**Table 1.** Descriptive statistics.

|  | Obs. | Min | Max | Mean | SD |
|---|---|---|---|---|---|
| *Donat* | 960 | 0.03 | 5.73 | 1.81 | 0.86 |
| *Idu* | 960 | 0.00 | 1.00 | 0.13 | 0.34 |
| *Disas* | 960 | 0.00 | 1.00 | 0.15 | 0.35 |
| *Economy* | 960 | 0.00 | 1.00 | 0.92 | 0.25 |
| *Operation* | 960 | 0.00 | 1.00 | 0.85 | 0.34 |
| *Harmony* | 960 | 0.00 | 1.00 | 0.07 | 0.26 |
| *Management* | 960 | 0.00 | 1.00 | 0.25 | 0.43 |
| *Size* | 960 | 8.14 | 12.26 | 9.84 | 0.60 |
| *Lev* | 960 | 0.00 | 0.89 | 0.40 | 0.20 |

Notes: *Donat* is the natural algorithm of the amount of charitable donation; *Idu* is a dummy variable, where 1 denotes the company belongs to one of the controversial industries and 0 denotes otherwise; *Disas* is a dummy variable for disaster year, where 1 denotes the observation is from a disaster year and 0 denotes the observation is not; *Economy* is a dummy variable for the SDGs of the profit maximization. It is proxied by ROE. This variable is 1 if ROE is positive, and it is 0 if otherwise; *Operation* is a dummy variable for the SDGs of the political interest. This variable is 1 if the company receives government subsidies, and this variable is 0 if otherwise; *Harmony* is a dummy variable for the SDGs of the altruistic. This variable is 1 if the company's net sales is negative, and this variable is 0 if otherwise; *Management* is a dummy variable for the SDGs of the managerial utility. In this variable, 1 denotes the CEO has no ownership of the firm, and 0 denotes the CEO has ownership of the firm; *Size* denotes firm size; *Lev* is debt ratio.

The correlations between the variables are shown in Table 2. Except for the correlation between *Size* and *Lev* (0.32) which is slightly higher, all the correlations between the variables are low. However, the variance inflation factors of *Size* and *Lev* are low (1.33 and 1.17 respectively). These results indicate no collinearity between the variables. In other words, each predictor has a unique characteristic, and none of the study variable is statistically similar to another.

**Table 2.** Correlation Matrix.

|  | Donat | Idu | Disas | Economy | Operation | Harmony | Management | Size | Lev |
|---|---|---|---|---|---|---|---|---|---|
| *Donat* | 1 | 0.00 | 0.11 | 0.08 *** | 0.10 *** | −0.02 | −0.08 ** | 0.30 *** | 0.09 *** |
| *Idu* |  | 1 | 0.02 | −0.01 | −0.06 * | −0.08 ** | −0.08 ** | −0.08 *** | −0.09 *** |
| *Disas* |  |  | 1 | −0.03 | 0.04 | −0.11 *** | −0.10 *** | −0.23 *** | 0.02 |
| *Economy* |  |  |  | 1 | 0.06 * | 0.08 ** | 0.07 ** | 0.01 | −0.15 *** |
| *Operation* |  |  |  |  | 1 | −0.05 | −0.09 *** | 0.04 | −0.04 |
| *Harmony* |  |  |  |  |  | 1 | 0.25 *** | 0.21 *** | 0.07 ** |
| *Management* |  |  |  | - |  |  | 1 | 0.10 *** | 0.04 |
| *Size* |  |  |  |  |  |  |  | 1 | 0.32 *** |
| *Lev* |  |  |  |  |  |  |  |  | 1 |

Notes: * $p < 0.1$; ** $p < 0.05$; *** $p < 0.01$.

*4.2. Regression Model Results*

Table 3 shows the firms' philanthropic giving behind the SDGs. It can be found that the SDGs of the economy, operation, and harmony are affected by philanthropic giving In other words, the sample firms are motivated by philanthropic giving to achieve the SDGs. In addition, philanthropic giving has a negative relation to managerial utility (*Management*). This suggests that a firm is less likely to engage in corporate philanthropy when its CEO has no ownership of the firm. The results also show that firm size (*Size*) is positively related to the SDGs. Probably due to more public attention, larger firms tend to attach greater importance to the SDGs. Finally, the relation between controversial industry (*Idu*) and the SDGs is not significant, but that between disaster period (*Disas*) and the SDGs is. As empirically pointed out in literature [42], Chinese firms donate more during a disaster period.

Further, this study examines whether the sample firms have a significant change in the SDGs due to industry characteristic or in the event of a disaster. The simultaneous

influence of each strategy combined with industry characteristic or disaster period is tested. The result is provided and discussed in the next section.

**Table 3.** Summary of results.

|  | H1 | | H2 | | H3 | | H4 | |
|---|---|---|---|---|---|---|---|---|
|  | **Coef.** | **Wald** | **Coef.** | **Wald** | **Coef.** | **Wald** | **Coef.** | **Wald** |
| *Donat* | 0.46 *** | 7.72 | 0.30 ** | 6.21 | 0.36 ** | 5.29 | −0.29 *** | 9.67 |
| *Idu* | −0.19 | 0.26 | −0.48 | 1.85 | −1.03 | 1.08 | −0.50 | 1.14 |
| *Disas* | 1.38 * | 3.46 | 0.35 ** | 4.14 | 1.99 * | 3.82 | 0.51 ** | 3.96 |
| *Size* | 0.19 | 0.62 | 0.25 | 1.82 | 1.32 *** | 31.30 | 0.45 *** | 10.07 |
| *Lev* | −3.45 *** | 23.31 | −1.06 ** | 4.67 | 0.40 *** | 44.27 | 0.11 | 0.08 |
| *Year effects* | Controlled | | Controlled | | Controlled | | Controlled | |
| *Obs.* | 960 | | 960 | | 960 | | 960 | |
| *Cox and Snell* | 0.76 | | 0.36 | | 0.53 | | 0.34 | |
| *Nagelkerke* | 0.89 | | 0.40 | | 0.62 | | 0.50 | |

Notes: * $p < 0.1$; ** $p < 0.05$; *** $p < 0.001$.

As shown in Table 4, all the interaction terms of the philanthropic strategy combined with industry characteristic and disaster period are not significant, meaning that the effects of philanthropic giving on the four SDGs is not significantly moderated by industry characteristic or disaster. Since the interaction terms are not significant, this study further tests the main effect of philanthropic giving. The negativity or positivity of each coefficient is the same as in Table 3. However, the coefficient for *Donat of H2* changes from 0.31 (see Panel A in Table 4) to 0.27 (see Panel B in Table 4). This manifests that the positive effect of philanthropic giving on the SDGs of economy is alleviated by industry characteristic or disaster, and industry characteristic also exhibits a stronger "sustainable operation effect" as compared with disaster. In other words, there will be a subtle change in sustainable operation of the company under the effect of industry characteristic and disaster. Moreover, firm size (*Size*) is positively related to the SDGs, no matter in the overall sample (Table 3) or when the moderation effect of industry characteristic and disaster is considered (Table 4). This finding confirms that larger firms pay more attention to the SDGs.

**Table 4.** Logistic regression analysis of the variables with consideration of the moderation of industry characteristic and disaster.

| **Panel A: The moderation of industry characteristic** | | | | | | | | |
|---|---|---|---|---|---|---|---|---|
|  | H1 | | H2 | | H3 | | H4 | |
|  | **Coef.** | **Wald** | **Coef.** | **Wald** | **Coef.** | **Wald** | **Coef.** | **Wald** |
| *Donat* | 0.35 ** | 4.28 | 0.31 ** | 5.86 | −0.37 ** | 5.31 | −0.13 ** | 3.01 |
| *Idu* [a] | −0.96 | 1.53 | −0.66 | 1.49 | −0.19 | 0.03 | 19.71 | 0.09 |
| *Donat × Idu* | 0.53 | 1.14 | 0.11 | 0.13 | −0.87 | 1.60 | −0.05 | 0.96 |
| *Size* | 0.28 | 1.40 | 0.18 | 1.11 | 1.47 *** | 39.12 | 2.40 ** | 3.76 |
| *Lev* | −3.51 *** | 24.23 | −0.99 ** | 4.16 | 0.20 | 0.08 | 1.88 | 2.57 |
| *Year effects* | Controlled | | Controlled | | Controlled | | Controlled | |
| *Obs.* | 960 | | 960 | | 960 | | 960 | |
| *Cox and Snell* | 0.80 | | 0.30 | | 0.57 | | 0.77 | |
| *Nagelkerke* | 0.91 | | 0.33 | | 0.93 | | 0.96 | |

**Table 4.** *Cont.*

| | Panel B: The moderation of disaster | | | | | | | |
|---|---|---|---|---|---|---|---|---|
| | H1 | | H2 | | H3 | | H4 | |
| | Coef. | Wald | Coef. | Wald | Coef. | Wald | Coef. | Wald |
| *Donat* | 0.48 *** | 7.16 | 0.27 ** | 4.84 | −0.08 | 0.07 | −0.35 ** | 4.96 |
| *Disas* | 0.39 ** | 5.08 | −0.87 * | 2.80 | 0.35 | 0.04 | 0.54 | 0.15 |
| *Donat × Disas* | −0.11 | 0.54 | 0.23 | 0.32 | −1.52 * | 1.52 | 0.13 | 0.04 |
| *Size* | 0.20 | 0.67 | 0.28 | 2.30 | 1.34 *** | 32.24 | −0.42 *** | 0.27 |
| *Lev* | −3.43 *** | 22.99 | −1.01 ** | 4.30 | 0.46 *** | 0.49 | 0.10 | 0.01 |
| *Year effects* | Controlled | | Controlled | | Controlled | | Controlled | |
| *Obs.* | 960 | | 960 | | 960 | | 960 | |
| *Cox and Snell* | 0.63 | | 0.30 | | 0.61 | | 0.50 | |
| *Nagelkerke* | 0.89 | | 0.38 | | 0.92 | | 0.52 | |

Notes: [a] *Idu* is a dummy variable where 1 indicates that firm belongs to any of the controversial industries and 0 indicates the firm does not; *Disas* is a dummy variable for year 2008 and year 2009 when China suffered a catastrophic earthquake and winter storms. 1 indicates the observation belongs to year 2008 or 2009, and 0 indicates the observation belongs to other years. * $p < 0.1$; ** $p < 0.05$; *** $p < 0.01$.

## 5. Discussion and Conclusions

In this paper, legitimacy theory is used as a foundation to explore Chinese firms' philanthropic giving for the SDGs and further analyze how philanthropic giving is related to the four aspects of SDGs, including the SDGs of profit maximization (*Economy*), political interest (*Operation*), altruistic (*Harmony*), and managerial utility (*Management*). Our empirical evidence shows that philanthropic giving has a positive effect on *Economy, Operation and Harmony*, and philanthropic giving has a negative relation with *Management*. This finding suggests that when the CEO has no ownership of the firm, the company would reduce its charitable donation and managerial utility and is even become less likely to engage in charitable giving. This study also probes into whether the sample firms have a change in their philanthropic strategy due to industry characteristic or in the event of a disaster. The empirical evidence indicates that the sample firms' philanthropic strategies do not significantly change by industry characteristic or the occurrence of a disaster. This explains that the sample firms' SDGs of economy, operation and harmony are mainly effected by philanthropic strategy. In other words, philanthropic giving is still the factor that enterprises in China first consider when making a decision about sustainable development goals. By making charitable donations, enterprises in China can establish corporate legitimacy and gain more subsidies from the government. From a different perspective, this also shows a common phenomenon in the contemporary Chinese society—enterprises prioritize their economic profit and neglect the traditional value that enterprises shall support "enterprise-run societies" (i.e., harmony) [16]. With corporate philanthropy, the motives for the SDGs have gradually evolved into the motives that are commonly seen in other countries. The findings of this study support legitimacy theory and also confirm that Chinese enterprises have motivations for the SDGs.

Moreover, the SDGs of the managerial utility is not common among the sample firms. This partly explains why corporate philanthropic behavior in China differs from that in Western countries [4]. For example, most enterprises in China are family enterprises. In most family enterprises, the CEO position is taken by a family member [43]. CEO's personal reputation in the human resources market is therefore not an important factor for family enterprises.

This study obtains insights that contribute to the practice and the literature. Previous research [15,20] argues that corporate philanthropy is a "window dressing" behavior or a symbolic behavior that is actually intended to evade social responsibilities. However, this study finds that under the motivations for legitimacy and corporate philanthropy, the sample firms would undertake some social responsibilities to seek the SDGs of economy, operation, and harmony. Business managers, supervisors, and decision-makers in countries with a similar social and economic system of China may find these findings interesting.

Finally, the SDGs of the managerial utility is not common among the Chinese firms. This finding highlights the motivations underlying the philanthropic behavior of enterprises in a socialist country undergoing economic reforms. This study finds that the positive effect of philanthropic giving on the SDGs of operation will be alleviated by disaster, and this "sustainable operation effect" is stronger in industry characteristic than in disaster. In other words, there will be a change in corporate operations when the company has such industry characteristic or when there is a major disaster.

This study has a number of limitations. First, some corporate attributes, such as capital structure and financial performance, would rise or decline frequently. The fluctuation of these attributes may have an effect on these firms' philanthropic decision. Therefore, to better explain these firms' philanthropic behavior, a time-series analysis is probably more suitable than a cross-sectional analysis. Second, recent studies [44] have argued that the effect of taxes should also be considered in the analysis of corporate philanthropy, because tax exemption encourages charitable donations. Since this study is unable to access data about tax savings from donations, the effect of taxes is not controlled for in the research model of this study.

**Author Contributions:** The authors who have contributed to this work are as follows: conceptualization, H.-C.Y. and L.K.; methodology, H.-C.Y. and L.K.; software, H.-C.Y.; validation, H.-C.Y. and L.K.; formal analysis, H.-C.Y.; investigation, H.-C.Y.; resources, H.-C.Y.; data curation, H.-C.Y.; writing—original draft preparation, H.-C.Y.; writing—review and editing, H.-C.Y. and L.K. All authors have read and agreed to the published version of the manuscript.

**Funding:** This research received no external funding.

**Data Availability Statement:** Not applicable.

**Acknowledgments:** The authors wish to thank the editor and the anonymous reviewers for constructive and insightful comments.

**Conflicts of Interest:** The authors declare no conflict of interest.

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
