# Peer review of "Corporate Philanthropy Strategy and Sustainable Development Goals"

_sustainability, doi:10.3390/su13105655_

Round 1
Reviewer 1 Report
It’s very interesting work looking at the strategic causes of charitable giving in the economic performance of companies in China.
However, at work there is no distinction between the activities of the companies, only some are excluded, or whatever their presence in the market. Perhaps one of the most decisive factors for the promotion and effect of the production of philanthropic donations by companies is precisely the factor that the authors have not been able to include in their work: tax benefits for charitable donations, which leaves the result of the work somewhat incomplete.
However, as a review of the content of the text make the following observations: statements such as those expressed in lines 34-37: «traditionally, it is better to keep a low profile when doing a good deed (e.g. making a charitable donation). Today, good deeds are publicly stressed and highlighted in the Chinese society... There is probably a social belief...» on the one hand they express opinions, not facts, and on the other hand they must be backed up by scientific arguments, especially when they are made before an analysis of results such as those that are subsequently presented. The same is true in line 117, when it is stated that «It is a widely agreed among enterprises that profit maximization is the true goal for giving». And in line 309, where it is stated that 'For example, except state-owned enterprises, most enterprises in China are family enterprises». That is certainly true, but it is a fact that is not contrasted with a statistical reference, which would be desirable.
Perhaps the authors should review the tense of verbs when formulating premises that are suggested as hypotheses of reality that may or may not occur.
Authors should be wary of, by using acronyms in the text, having initially noticed the content of the acronyms that form it. Many are well known, but even these should be explicit.
The statement in line 90: «meanwhile, this work will be one of the key factors that will affect sustainable development goals», is pretentious and I think it is not convenient.
The final statements made by the authors in each paragraph dedicated to «Goals for Corporate Charitable strategy», lines 128, 141, 150 and 166, are as categorical as lack of justification: how?, to what extent?...
The authors start with the analysis of «Literature Review» of the «Legitimacy theory perspective», but without offering a definition or concept of the content of the theory, with which the exhibition can be difficult to follow. The same applies to the «perspective of agency theory» (line 151).
Author Response
Dear Reviewer
We have revised it as follows file"sustainability-1212652 R1. (reviewer 1)
Please see the attachment.

Reviewer 2 Report
The authors of the article conducted a survey about charitable giving strategy in China, it is an interesting examination based on wide secondary data, with some minor problems as noticed in the list below.
Important overall notice: the title should be reconsidered, as the article discusses Chinese firms’ strategic motives for corporate philanthropy. The term "charitable giving" is mentioned four times, (1 is in references).
As an overall opinion, the research goals shall be more highlighted in Introduction. It should be highligted why the used database was chosen, and some details about this database.
- Abbreviation of Sustainable Developme Goals is SDGs, it should be used consequently.
- In row 28, first part of the sentence is missing. Next sentence "we will..." is unclear.
- It is suggested to avouid using "we" it is better to refer as "authors" or to use simply passive voice.
- Research objectives (from row 50) are not highlighted. They should be separated in paragraphs at least, but a figure or table that summarizes the research concept is also recommended.
- Row. 68: ..previous research shows.... Source? Which research??
- Row 88: POEs - full term for abbreviations shall be given at first mentioning.
- Row 106, subchapter entitled Goals for Corporate Charitable Strategy... it includes Hypotheses, which is not for Literature review... please reconsider it.
- Row 113 and 114: space before comma (a comma is at the beginning of the next row),
- Rows 116, 129, 142, 151: First sentence is a fragment (it seems to be a subtitle, please check&correct)
- Row 137: (p.180)????
- Row 170: CSMAR database? full name shall be mentioned, this database is not mentioned in References section
- Row 170: 2008~2015 should be 2008-2015
- Row 175: first sentence is fragmented
- Row 178: SDG is in italic
- Row 180: ROE - full term for abbreviations shall be given at first mentioning.
- Subchapter. 3.2. - The research model should be written more clearly. Variables should be separated in paragraphs to be more transparent. A figure showing the model could also be recommended.
- Table 4, please check formattin of head rows (in bold)
- Authors' contribution: please use initials for the name or fullname, but consequently
- Funding, Data, Acknowledgement: shall be checked and modified.
Author Response
Dear Reviewer
We have revised it, as follows file"sustainability-1212652 R1. (reviewer 1).
Please see the attachment.

Round 2
Reviewer 1 Report
Thanks for the effort in the review. Lucky